# Assessing the impact of the National Department of Health's National Adherence Guidelines for Chronic Diseases in South Africa using routinely collected data: a cluster-randomised evaluation

Matthew P Fox,[1,2,3] Sophie J Pascoe,[2] Amy N Huber,[2] Joshua Murphy,[2] Mokgadi Phokojoe,[4] Marelize Gorgens,[5] Sydney Rosen,[1,2] David Wilson,[5] Yogan Pillay,[4] Nicole Fraser-Hurt[5]

For numbered affiliations see end of article.

**Correspondence to**
Dr Matthew P Fox;
mfox@bu.edu

## ABSTRACT

**Introduction** In 2016, South Africa's National Department of Health (NDOH) launched the National Adherence Guidelines for Chronic Diseases for phased implementation throughout South Africa. Early implementation of a 'minimum package' of eight interventions in the Adherence Guidelines for patients with HIV is being undertaken at 12 primary health clinics and community health centres in four provinces. NDOH and its partners are evaluating the impact of five of the interventions in four provinces in South Africa.

**Methods and analysis** The minimum package is being delivered at the 12 health facilities under NDOH guidance and through local health authorities. The five evaluation interventions are: (1) fast track initiation counselling for patients eligible for antiretroviral therapy (ART); (2) adherence clubs for stable ART patients; (3) decentralised medication delivery for stable ART patients; (4) enhanced adherence counselling for unstable ART patients; and (5) early tracing of patients who miss an appointment by ≥5 days. For evaluation, NDOH matched the 12 intervention clinics with 12 comparison clinics and randomly allocated one member of each pair to intervention or comparison (standard of care) status within pairs, allowing evaluation of the interventions using a matched cluster-randomised design. The evaluation uses data routinely collected by the clinics, with no study interaction with subjects to prevent influencing the primary outcomes. Enrolment began on 20 June 2016 and was completed on 16 December 2016. A total of 3456 patients were enrolled and will now be followed for 14 months to estimate effects on short-term and final outcomes. Primary outcomes include viral suppression, retention and medication pickups, evaluated at two time points during follow-up.

**Ethics and dissemination** The study received approval from the University of Witwatersrand Human Research Ethics Committee and Boston University Institutional Review Board. Results will be presented to key stakeholders and at international conferences and published in peer-reviewed journals.

### Strengths and limitations of this study

- ► The evaluation assesses the impact of an adherence strategy that is planned for national implementation to improve adherence and retention and decongest clinics.
- ► The evaluation allows for rigorous evaluation and improvement of the interventions by using a randomised roll-out by the government with minimal burden on healthcare workers and patients.
- ► However, as we do not control the implementation of the interventions, we cannot ensure that they are followed according to guidelines. Further, if patients self-select into the interventions, this could also create some selection bias.
- ► In addition, as we do not control the data collection in the clinical files, we do have some missing and misclassified data and lack of blinding could lead to some misclassification.
- ► Finally, as many of these interventions are improvements on previous approaches that are already part of guidelines, we do not have pure control group.

**Trial registration number** NCT02536768; Pre-results.

## INTRODUCTION

For antiretroviral therapy (ART) for HIV to be effective, patients must remain in care for long periods of time, initiate treatment as early as allowed under prevailing guidelines (now typically immediately after diagnosis in many countries), remain in care, consistently achieve high levels of adherence to their treatment regimen and, as a result, achieve and sustain an undetectable viral load. Treatment is lifelong and requires consistent,

nearly complete daily adherence to be successful. In South Africa, where some 3.7 million individuals are now on ART,[1] numerous studies and reviews[2–5] and the South African National Department of Health's (NDOH) own data[6] have indicated that retention in care and adherence to ART in South Africa are suboptimal and pose a serious threat to the long-term success of the national HIV response.

To address this challenge, in 2014 the NDOH developed the 'National Adherence Guidelines for Chronic Diseases (HIV, TB and NCDs)'.[7 8] The guidelines call for the provision of a minimum package of eight interventions to increase linkage to care, retention in care and adherence to treatment. Although there is some published and unpublished evidence of the effectiveness of each of these interventions for HIV care,[9–11] most have not been implemented jointly or at scale, nor have they been evaluated as delivered routinely by public sector facilities, without external technical assistance or resource support. Better information is needed to guide the NDOH's roll-out of the minimum package at national scale and about the number of patients requiring each intervention.

Prior to national scale-up of the Adherence Guidelines, the NDOH selected 12 clinics (primary healthcare clinics (PHC) and community health centres) for early implementation of the minimum package for patients with HIV. This will generate information to refine the guidelines and gain experience in implementation. This manuscript presents the protocol for a matched cluster-randomised evaluation to assess the impact of five of the interventions that are part of the National Adherence Guidelines on HIV retention and viral suppression outcomes at public sector clinics. The specific objectives of the evaluation are to evaluate the impact of:

1. Among HIV-infected patients newly eligible for ART, *fast track treatment initiation counselling* on ART initiation and viral suppression.
2. Among HIV-infected patients who are stable on ART, *adherence clubs* (AC) on ART adherence and viral suppression.
3. Among HIV-infected patients who are stable on ART, *decentralised medication delivery* (DMD) on ART adherence and viral suppression.
4. Among HIV-infected patients who have poor adherence (as indicated by an unsuppressed viral load) to ART, *enhanced adherence counselling* (EAC) on ART adherence and viral suppression.
5. Among HIV-infected patients in ART programmes who miss a scheduled appointment by 5 days, *early patient tracing* on retention in care.

## METHODS AND ANALYSIS
### Study design
The study will estimate the effectiveness of each of five interventions included in the minimum package of adherence interventions using a matched cluster-randomised design. It is taking place at 24 public sector clinics in

**Table 1** South Africa's National Adherence Guidelines minimum package of interventions

| Approach | Intervention |
| --- | --- |
| Education and counselling | 1. Fast track initiation counselling* <br> 2. Enhanced adherence counselling for unstable patients* <br> 3. Child disclosure counselling for children living with HIV |
| Repeat prescription collection strategies | 4. Adherence clubs* <br> 5. Spaced and fast lane appointment systems <br> 6. Decentralised medication delivery* |
| Patient tracing | 7. Early tracing of all missed appointments* |
| Integrated HIV, TB, NCD care | 8. Integrated consultation and counselling |

*Indicates interventions included in this evaluation.
NCD, non-communicable disease; TB, tuberculosis.

four provinces in South Africa. The interventions were developed by South Africa's NDOH and are being implemented by the clinics using their own staff and resources; the study itself is providing no additional services. All non-pregnant adult patients seeking HIV-related services at the study sites and eligible to receive one of the interventions during the study enrolment period are eligible for inclusion in the study. The study has no direct interaction with study subjects and no study visits. Data are instead collected from routinely completed patient records, including clinic files, registers and databases. Because the roll-out was conducted by the NDOH and the interventions delivered by the sites, the study team did not have contact with individual patients. Instead, the study was approved for analysis of data routinely collected by the study sites. As no patient contact occurred, we received a waiver of consent.

### Interventions
The interventions in the minimum package are listed in table 1. The following five interventions will be evaluated under this protocol. Each is described in detail in the National Adherence Guidelines, which are available at https://www.nacosa.org.za/wp-content/uploads/2016/11/Integrated-Adherence-Guidelines-NDOH.pdf.[7]

1. *Fast track initiation counselling (FTIC)* seeks to reduce attrition from chronic care by speeding up the process of treatment initiation for patients who are eligible for treatment and thereby increasing the proportion of treatment-eligible patients who start treatment promptly. For HIV, the goal is to reduce the total number of visits that patients need to complete in order to start treatment and allow patients to initiate treatment over the course of two clinic visits within 1 week of confirming ART eligibility, with additional counselling provided in the first two routine visits after treatment initiation. The intervention includes a detailed curriculum for the counselling sessions, and

providers work with patients to create an individualised adherence plan and ensure postinitiation adherence support.[12]

2. *ACs* comprise adherent and stable patients on ART who meet at facilities or identified locations in the community, in groups of up to 30 patients every 2–3 months to receive group counselling, have a clinical assessment and receive the required supply of prepacked medications. ACs are facilitated by a nurse and lay staff at the healthcare facility with support from community health workers. The goal is to keep patients engaged in care and adherent to their medication by providing social support and facilitating medication delivery and treatment monitoring, while also reducing patient visit burden on the clinics.[13] The adherence guidelines' standard operating procedure provides detailed instructions for establishing and running the clubs and for eligibility criteria and data collection.

3. *DMD* through the Central Chronic Medicine Dispensing and Distribution and Chronic Dispensing Unit programmes uses locations other than the clinic pharmacy to deliver medications to patients who are stable on treatment. Patients then only need to come to the clinic on a six monthly basis for a clinical exam. The goal is to reduce the burden on the patient in terms of the time and resources it takes them to collect their medication to improve treatment adherence and retention in care, while also decongesting the clinics.

4. *EAC* seeks to identify patients who have poor treatment adherence as indicated by an elevated viral load and target these patients for EAC to help them improve their adherence. Although HIV-infected patients with detectable viral loads may receive additional adherence counselling under the standard of care, this intervention will standardise and intensify that counselling. The intervention includes one or two structured education/counselling sessions in which effective strategies for achieving good adherence are discussed and goals set for viral resuppression.

5. *Early tracing and retention in care (TRIC) of patients* who miss an appointment by 5 days or more seeks to identify patients who have not returned to the clinic for scheduled appointments and attempts to return them to care through contact by phones, text message and/or home visits. This intervention requires obtaining permission from patients to contact them and maintaining up-to-date contact information in patient records. The goal is to reduce clinic loss to follow-up and improve patient outcomes by identifying those who have missed appointments and encouraging them to return to care.

Each of the interventions, while implemented as a package of services, is delivered to a unique population within each clinic: patients newly eligible for ART (FTIC), patients stable on ART (DMD or AC), patients with poor adherence (EAC) and patients lost from care (TRIC). Patients who are stable on ART can be provided with either DMD or ACs, but not both. These two interventions were offered to stable patients in an effort to provide patients with multiple options for where and how to seek care while reducing the congestion in clinics and increasing the convenience for patients. Because the patient populations differ, the study can estimate the effect of each intervention individually, in the context of implementation of the overall minimum package.

In order for the study to be as close as possible to routine care conditions, the interventions for this study are being implemented by the study sites with no input or oversight from the study team. The interventions follow the National Adherence Guidelines and NDOH has organised trainings prior to the implementation of the interventions to support appropriate implementation of the guidelines.

### Selection and randomisation of study sites
The evaluation is being conducted at 24 PHCs in South Africa. All study sites follow the current guidelines for HIV care and treatment, dated December 2014.[14] Six clinics were chosen from one district each in Gauteng, KwaZulu-Natal, Limpopo and North West Provinces. These provinces were chosen in consultation with NDOH to represent high HIV burden regions with high-burden districts and high-volume clinics. The study team developed a list of all sites in each participating province that met these criteria and selected three matched pairs of clinics per province. Pairs were matched on ART patient volume (1000–1999, 2000–4999, or ≥5000 current ART patients), setting (urban, informal settlement, or rural), location (pairs should be located relatively nearby one another) and HIV viral suppression rate (see table 2). In each pair, one clinic was randomly assigned (using a computer-generated randomisation) to receive early implementation of the minimum package of interventions, while the other continued to provide standard of care. No blinding was used.

### Inclusion and exclusion criteria
For each objective, we enrolled a specific cohort of patients as shown in figure 1. All cohorts included patients aged 18 years or above and excluded patients who are not resident in the facility's catchment area, were recorded as having an intention to transfer care to a different facility within 12 months, or were pregnant and eligible for prevention of mother to child transmission (PMTCT). Each cohort had specific inclusion criteria related to eligibility for the specific intervention. These criteria followed the December 2014 national guidelines for HIV care and ART[14] and July 2016 National Adherence Guidelines for Chronic Disease (HIV, tuberculosis and non-communicable diseases).[11] In order to identify eligible patients to enrol, we first identified all patients eligible for each intervention based on information recorded on their electronic medical record. At intervention sites, lists were reviewed against clinic records, registers and other documentation for each intervention to identify eligible patients. At control sites, we reviewed lists

**Table 2** Location of early learning sites, allocation status and value of matching variables used to determine matched pairs[15]

| District and province | Pair | Site number | Subdistrict * | Study allocation | Total remaining on ART (November 2014) | % of viral loads where VL <400 copies/mL† |
|---|---|---|---|---|---|---|
| Ekurhuleni, Gauteng | 1 | 1 | S2 | Intervention | 1094 | 71 |
| | | 2 | S2 | Control | 1098 | 66 |
| | 2 | 3 | S2 | Intervention | 2676 | 86 |
| | | 4 | S2 | Control | 2749 | 76 |
| | 3 | 5 | S2 | Intervention | 1929 | 84 |
| | | 6 | S2 | Control | 1072 | 80 |
| Mopani, Limpopo | 4 | 7 | Greater Tzaneen | Intervention | 1720 | 71 |
| | | 8 | Greater Tzaneen | Control | 1022 | 64 |
| | 5 | 9 | Greater Giyani | Intervention | 1702 | 73 |
| | | 10 | Greater Giyani | Control | 1445 | 76 |
| | 6 | 11 | Greater Tzaneen | Intervention | 1370 | 77 |
| | | 12 | Greater Tzaneen | Control | 1027 | 76 |
| Bojanala Platinum, North West | 7 | 13 | Madibeng | Intervention | 4147 | 83 |
| | | 14 | Madibeng | Control | 4182 | 82 |
| | 8 | 15 | Madibeng | Intervention | 1152 | 83 |
| | | 16 | Madibeng | Control | 1224 | 81 |
| | 9 | 17 | Rustenburg | Intervention | 3951 | 80 |
| | | 18 | Rustenburg | Control | 3328 | 78 |
| King Cetshwayo (previously uThungulu), KwaZulu-Natal | 10 | 19 | uMlalazi | Intervention | 1900 | 72 |
| | | 20 | uMlalazi | Control | 1053 | 69 |
| | 11 | 21 | uMhlathuze | Intervention | 5037 | 83 |
| | | 22 | uMhlathuze | Control | 7305 | 82 |
| | 12 | 23 | Ntambanana | Intervention | 1111 | 81 |
| | | 24 | Ntambanana | Control | 1184 | 88 |

*Used as proxy for setting and location.
†NHLS data April 2014 to March 2015.
ART, antiretroviral therapy; NHLS, National Health Laboratory Service; VL, viral load.

against clinic records to confirm eligibility. If the patient file was found and eligibility for a cohort was confirmed then patients were enrolled sequentially until the required sample size was reached for that cohort. Due to delays in electronic data capturing data was not complete. To account for this at some sites, individuals receiving each intervention were identified directly from registers for that intervention. Clinic files were then reviewed to confirm eligibility and patients were enrolled until the required sample size was achieved. For each patient enrolled, regardless of the method used to identify them, patient files were reviewed and information was extracted using an electronic case report form to confirm patients met all eligibility criteria for that cohort.

## Duration of follow-up
The study enrolment period began on 20 June 2016 and was completed on 16 December 2016. For individuals, observation began on the date of determination of eligibility for an intervention. Follow-up of the cohorts is now ongoing and is anticipated to be completed in December 2017. Passive follow-up through medical record and database review will continue for a minimum of 14 months after the date of enrolment (2 additional months beyond 12 months to allow 1-year outcomes to occur and be recorded). This will allow all subjects sufficient follow-up time to complete each of the primary outcomes designated.

## Data sources
As noted, this study is relying on routinely collected data for study outcomes. Routine data sources will include TIER.Net, the National Health Laboratory Service (NHLS) database which contains all laboratory tests done in public sector clinics, and data sets created by entering information from clinic registers, adherence plans and patient clinic files into a database. Various degrees of strengthening of existing data collection procedures were needed at the facilities in order to ensure complete entries into existing clinic registers or patient files, complete and

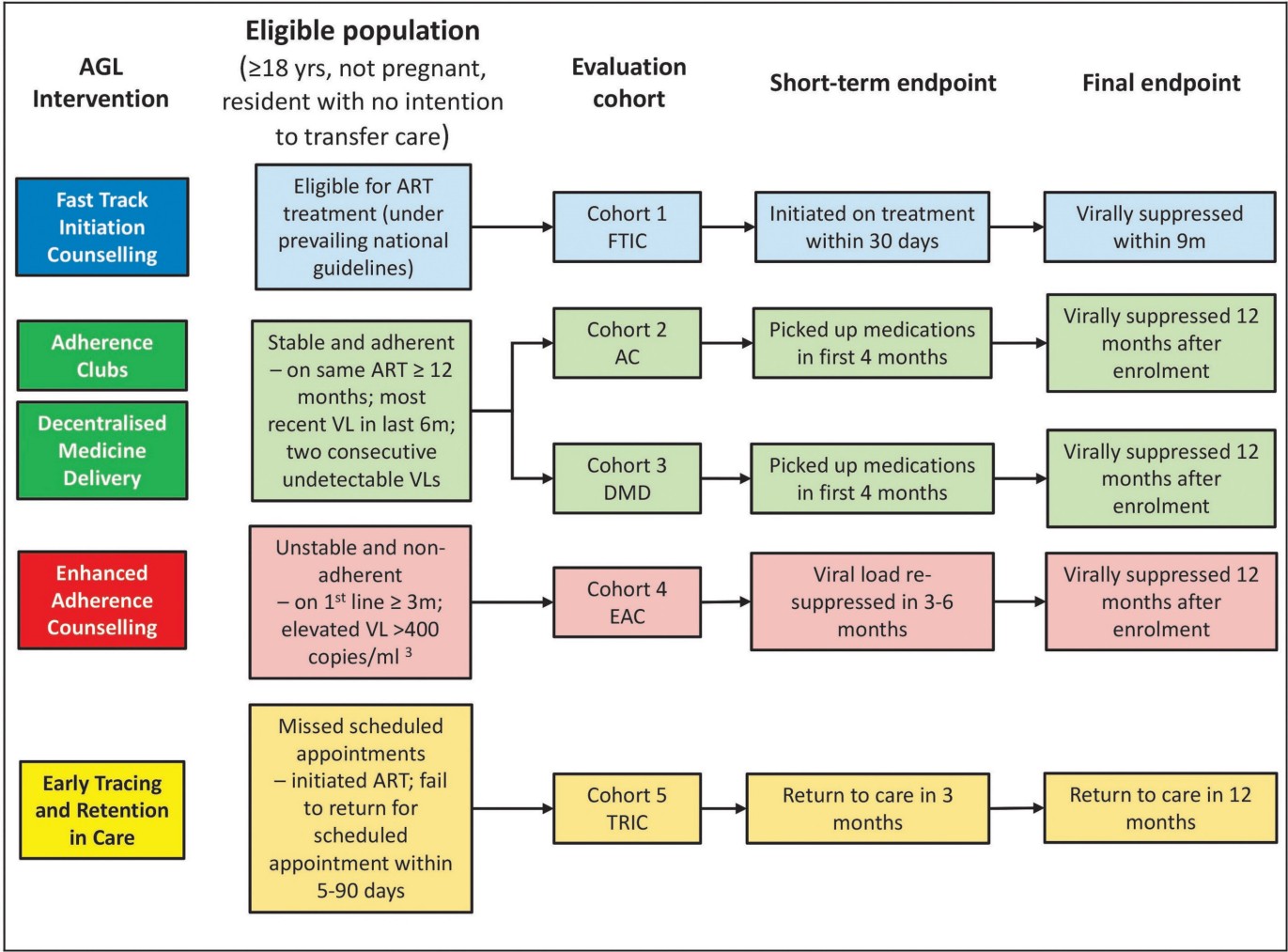

**Figure 1** Eligible population for each evaluation cohort and short-term and final endpoints for the impact evaluation. AC, adherence club; AGL, Adherence Guidelines; ART, antiretroviral therapy; DMD, decentralised medication delivery; EAC, enhanced adherence counselling; FTIC, fast track initiation counselling; TRIC, early tracing and retention in care; VL, viral load.

accurate entry of source data onto electronic files and the use of a consistent clinic-level patient identifier to link patients between data sources (eg, a register containing a row for each visit and a patient file containing documents pertaining to that patient will each contribute to the evaluation record for that patient).

## Study outcomes

Table 3 and figure 1 list the primary and secondary outcomes we will measure for each of the objectives. Each primary outcome includes both a short-term (S) outcome and a final (F) outcome for assessment of the immediate and longer term effects of the intervention. Short-term outcomes are typically focused on retention-based outcomes within the first 3–4 months after the intervention, with the exception of FTIC in which we assess the impact on treatment initiation within the first month after eligibility. Final outcomes are focused mainly on retention and viral suppression at 12 months. Note that viral suppression in the current South African ART guidelines is defined as viral load below 400 copies/mL[3], and this is the threshold that South Africa's NHLS reports.

For all outcomes, retention in care is defined as (1-%attrition), with attrition calculated as the sum of reported deaths, loss to follow-up and reported transfers to other facilities. Retention will thus be interpreted as 'retained in care at facility,' since the outcomes of patients who transfer will not be known. Loss to follow-up is defined as failure to attend the clinic within 90 days of a scheduled appointment, as stated in the Adherence Guidelines.

## Sample size

Table 4 shows the sample size that is required to detect meaningful differences for objectives 1–5. Sample sizes were determined using PASS software for cluster-randomised designs. Each sample size was determined to measure our short-term outcome for the objective. All calculations assume a site-clustered design with the clinic as the cluster and 24 clusters evenly split between intervention and comparison groups. We assumed power of 80% and an alpha of 0.05. Sample sizes accounted for the cluster-randomised design by assuming a coefficient of variation of 0.1. Each sample size was calculated assuming

**Table 3** Short-term (S) and final (F) evaluation outcomes for the Adherence Guideline impact evaluation in South Africa

| Objective | Primary outcome | Secondary outcomes |
|---|---|---|
| Fast track ART initiation counselling (objective/cohort 1) | Proportion of patients who initiate ART within 30 days of becoming ART eligible (S) and the proportion of patients who are alive, in care and virally suppressed (<400 copies/mL$^3$) within 9 months of ART eligibility (F) | Proportion of patients who initiate ART within 1 week of becoming ART eligible Demographic and clinical characteristics of patients who do and do not achieve primary outcomes (age, sex, baseline CD4 counts, TB diagnosis, other characteristics as allowed by data) |
| Adherence clubs (objective/cohort 2) | Proportion of patients eligible for participation in an adherence club who receive all medications within the first 3 months after club eligibility (S) and the proportion virally suppressed (<400 copies/mL$^3$) at 12 months after club eligibility (F) | Proportion of patients consistently participating in a club Demographic and clinical characteristics of patients who do and do not achieve primary outcomes |
| Decentralised medication delivery (objective/cohort 3) | Proportion of patients eligible for decentralised medication delivery who receive all medications within the first 3 (S) months after delivery eligibility and viral suppression (<400 copies/mL$^3$) 12 months after delivery eligibility (F) | Proportion of patients consistently receiving medications Demographic and clinical characteristics of patients who do and do not achieve primary outcomes |
| Enhanced adherence counselling (objective/cohort 4) | Proportion of patients with an elevated viral load who are alive, retained in care and resuppress their viral load (<400 copies/mL$^3$) within 3 (S) and 12 months (F) of eligibility for enhanced adherence counselling | Demographic and clinical characteristics of patients who do and do not achieve primary outcomes |
| Early tracing of patients lost to follow-up (objective/cohort 5) | Proportion of patients eligible for early patient tracing who return to care within 3 (S) and 12 (F) months of eligibility | Proportion of patients reached by tracers Number of tracing attempts required; proportion of patients retained in care for at least one additional routine visit after tracing Demographic and clinical characteristics of patients who do and do not achieve primary outcomes |

ART, antiretroviral therapy; TB, tuberculosis.

a baseline proportion of patients achieving the outcome in the absence of the intervention as determined from the literature or experience. Sample sizes were calculated based on being able to detect an absolute increase on outcomes deemed to be clinically meaningful, ranging from 15% to 20% as determined by consensus of the investigators. The total sample size was calculated to be 3456 including all of the five HIV cohorts.

## Data analysis

Our general analytical plan will be the same for each of the five interventions. We will begin with descriptive analyses of the characteristics of each of the cohorts stratified by intervention/non-intervention (comparison) status. We will also look for differences within the randomised matched pairs. Because the data will be collected as part of a clustered design, the data analysis will need to account for clustering. For each primary outcome described above, we will conduct a crude analysis comparing the proportion of subjects with the outcome in the intervention and comparison arms. Next, we will conduct an analysis for each outcome accounting only for clustering using generalised estimating equations (GEE) with an unstructured correlation matrix and clustering by treatment site. In all cases, the outcomes are dichotomous and therefore we will calculate relative

risks or risk differences comparing the intervention with the comparison arms using a log (or identify) link function and a binomial distribution and will adjust for matched pairs. Next, should any imbalances between treatment groups be detected, we will adjust for those covariates in our GEE model using covariate adjustment. We will look for differences in the effects of the strategies by important baseline characteristics (eg, size of the treatment population, rural vs urban, province, and so on) using stratified analyses. In addition, as we are using routine data collection for this study and because much of the outcome data are in electronic data going back to before the period of the intervention, we will also be able to adjust for baseline imbalances using difference-in-differences analyses.

## Ethics and dissemination

The study has received ethics approval from both the University of the Witwatersrand Human Research Ethics Committee (Medical) and the Boston University Institutional Review Board. In South Africa, we have also received national, provincial and district-level approvals and the trial has been registered at ClinicalTrials.gov (NCT02536768). Results of the study will be presented to key stakeholders as well as at international conferences and published in peer-reviewed journals.

**Table 4** Sample sizes for each objective of the Adherence Guideline impact evaluation study in South Africa

| Objective | Sample size | Rationale |
|---|---|---|
| Objective 1—fast track ART initiation counselling | 720 patients | The RapIT study of rapid ART initiation,[16] conducted at a well-managed PHC in Gauteng Province, found that about 60% of ART-eligible patients initiated under standard care within 30 days. Conservatively assuming 60% initiation without the intervention and 75% with the intervention, 30 subjects in each of the 24 clusters for 720 total subjects will be required to detect a difference of 15%. We have increased this by 20% to account for ineligible patients. |
| Objective 2—adherence clubs | 576 patients | Data from Themba Lethu Clinic[17–19] show that about 80% of patients made all of their medication pickups over a 3-month period. It is anticipated that 24 subjects per clinic for a total of 576 patients will be needed to detect a difference of 15%. We have increased this by 20% to account for ineligible patients. |
| Objective 3—decentralised medication delivery | 576 patients | Data from Themba Lethu Clinic[17–19] show about 80% of patients made all of their medication pickups over a 3-month period. It is anticipated that 24 subjects per clinic for a total of 576 patients will be needed to detect a difference of 15%. We have increased this by 20% to account for ineligible patients. |
| Objective 4—enhanced adherence counselling | 1008 patients | Data from KwaZulu-Natal Province indicate that 52% of patients with a detectable viral load resuppress after one session. It is anticipated that 42 subjects per clinic for a total of 1008 patients will be needed to detect a difference of 15%. We have increased this by 20% to account for ineligible patients. |
| Objective 5—early tracing of patients lost to follow-up | 576 patients | Data from various Right to Care clinics[19] suggest that the proportion of patients who are lost from care who return with no or little intervention is low, between 20% and 35%. It is anticipated that 24 subjects per clinic for a total of 576 patients will be needed to detect a difference of 15% assuming a baseline of 30% loss to follow-up without intervention. We have increased this by 20% to account for ineligible patients. |

ART, antiretroviral therapy; PHC, primary healthcare clinic; RapIT, Rapid Initiation of Treatment Trial.

The study team does not have any interaction with study subjects. Data for the study are drawn from existing records that are routinely collected at the study sites as part of routine patient care. We therefore believe that our study poses no physical risks to subjects. The only risk that we believe is posed by this study is that of loss of confidentiality. We are collecting data indicating individuals' HIV status and other sensitive health information. A high level of stigmatisation continues to inhibit the disclosure of HIV status in the study population. A breach of confidentiality, for example, through inadvertent loss of a storage device or paper files, would thus pose a risk to subjects.

We are protecting against the risk and repercussions of loss of confidentiality in two main ways. First, patient identifiers are stored separately from all other individual data in encrypted, password-protected files. Analytical data sets will not contain any identifiers, and the linking files containing the identifiers will be destroyed once all linking has been accomplished. Second, all study data are stored in secure locations. Password-protected laptops and tablets used on-site are kept in locked and secure locations when not in use. All data collected on tablets are immediately uploaded to a secure cloud server as soon as data collection for a patient is complete and is not kept on the tablets. Patient data extracted from electronic patient systems are extracted in a password-protected double-encrypted format and uploaded to a secure server via a dedicated secure virtual private network. All study staff have been trained in Good

Clinical Practice, Research Ethics and study procedures to ensure that they understand both research confidentiality requirements and study confidentiality procedures. Study investigators monitor data collection on an ongoing basis.

We are not seeking informed consent for this study, which is a record review only and poses minimal risk to study subjects. The interventions have been provided by the study clinics as standard care under the early roll-out of NDOH's new Adherence Guidelines, not as part of the study itself.

### Dissemination of findings

The primary audience for this evaluation is the South African NDOH and its partners, which will use the results to improve, target and budget for the national implementation of the Adherence Guidelines. Many of the findings, however, will likely be of broader interest in South Africa and other countries, where effective strategies for improving chronic disease medication adherence are eagerly sought. Results of the evaluation will be made as widely available as possible through journals, websites and conferences. Only aggregated, stratified data will be presented and it will not be possible to identify any individual patients from any of the data that are presented.

**Author affiliations**
[1]Department of Global Health, Boston University School of Public Health, Boston, Massachusetts, USA

[2] Health Economics and Epidemiology Research Office, Department of Internal Medicine, School of Clinical Medicine, Faculty of Health Sciences, University of the Witwatersrand, Johannesburg, South Africa

[3] Department of Epidemiology, Boston University School of Public Health, Boston, Massachusetts, USA

[4] National Department of Health, Pretoria, South Africa

[5] The World Bank Group, Washington, DC, USA

**Contributors** NFH, MG, MP, SP, SR and MPF all contributed to developing the protocol. AH, JM, DW, MG and YP all contributed substantive changes to the protocol. MPF drafted the manuscript. All authors were involved in editing the final manuscript.

**Funding** This work was supported by the World Bank trust funds from several governments and Government of South Africa domestic health financing.

**Competing interests** None declared.

**Patient consent** Not required.

**Ethics approval** University of Witwatersrand Human Research Ethics Committee and Boston University Institutional Review Board.

**Provenance and peer review** Not commissioned; externally peer reviewed.

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
