## [Reviewer comments · BMJ Open]

ARTICLE DETAILS

TITLE (PROVISIONAL)	Assessing the Impact of the National Department of Health's National Adherence Guidelines for Chronic Diseases in South Africa Using Routinely Collected Data: A Cluster-Randomised Evaluation
AUTHORS	Fox, Matt; Pascoe, Sophie; Huber, Amy; Murphy, Josh; Phokojoe, Mokgadi; Gorgens, Marelize; Rosen, Sydney; Wilson, David; Pillay, Yogan; Fraser-Hurt, Nicole

VERSION 1 – REVIEW

REVIEWER	Becky Genberg Johns Hopkins Bloomberg School of Public Health, USA
REVIEW RETURNED	11-Oct-2017

GENERAL COMMENTS	This paper clearly articulates a study protocol of a cluster-randomized trial to evaluate the impact of five interventions to promote HIV retention in care and viral suppression as part of the implementation of the National Adherence Guidelines for Chronic Diseases in South Africa. The methods for the study design and data analysis plans are thoroughly described. A few questions for clarification on the interventions for stable patients. Might self-selection into one intervention or the other impact the results of the evaluation? While the randomization occurs at the level of the facility, the analysis is being conducted at the patient level, and there may be imbalance in the study groups due to factors associated with choosing clubs vs. decentralized delivery. How is this being accounted for in the analysis plan? Also, why are clubs being evaluated with a 3 month timeframe while decentralization uses a 4-month timeframe? Additional information to understand the rationale for the evaluation of these two interventions may be useful.
---

REVIEWER	Reed Siemieniuk McMaster University, Canada
REVIEW RETURNED	18-Oct-2017

GENERAL COMMENTS	Thank you for the opportunity to peer review this manuscript. The authors describe their matched-cluster randomised trial that evaluates five different social interventions to improve adherence among people living with HIV in South Africa. Congratulations on the major effort to address important implementation science questions.
--

	It is obscure to me why the authors have decided to submit the protocol so late. The study has already finished enrolling and follow-up is already finished for most of the patients. There are so few clusters that there is very likely going to baseline imbalance. The results, whatever they happen to be, are going to be at high risk of bias. The authors refer to 12-14 month follow-up as “long-term”. This is semantics, but no one who is living with HIV indefinitely would consider 1-year follow-up to be long-term or even medium term. I suggest a change in terminology. Were the people who reviewed patients for eligibility blinded, or was anything else done to reduce patient selection bias? Why did the authors assume a coefficient of variation (ICC) of 0.1? This can be very difficult to predict and should be justified. Was the same ICC used for every analysis? Was any sensitivity analysis performed across a range of ICCs? The study sites were matched. The analysis plan does not currently incorporate this might be more reliable (not to mention increase power) if they do. The authors say that changes in the outcomes of 10% to 15% are "clinically meaningful" and imply that changes less than 10-15% are not clinically meaningful. How did they decide this threshold?
--	---

VERSION 1 – AUTHOR RESPONSE

Reviewer: 1

Reviewer Name: Becky Genberg

Institution and Country: Johns Hopkins Bloomberg School of Public Health, USA Please state any competing interests: None declared

Please leave your comments for the authors below

Comment: This paper clearly articulates a study protocol of a cluster-randomized trial to evaluate the impact of five interventions to promote HIV retention in care and viral suppression as part of the implementation of the National Adherence Guidelines for Chronic Diseases in South Africa. The methods for the study design and data analysis plans are thoroughly described.

Response: We thank you for taking the time to review our paper.

Comment: A few questions for clarification on the interventions for stable patients. Might self-selection into one intervention or the other impact the results of the evaluation?

Response: You raise a good point. This should not be true as all eligible subjects in the study clinics that are implementing the interventions are supposed to get the interventions, however we suspect that this is not the case. As such, we have added this as a limitation.

Comment: While the randomization occurs at the level of the facility, the analysis is being conducted at the patient level, and there may be imbalance in the study groups due to factors associated with choosing clubs vs. decentralized delivery. How is this being accounted for in the analysis plan?

Response: This is also an important point. While we initially only planned to deal with this through simple statistical adjustment, we have since realized that we will have sufficient site level data to deal with some of this through a difference in differences approach to adjust for baseline imbalances in outcomes. This is because, since we are using routinely collected data, some of which is collected in electronic format, we will be able to identify outcomes for the entire clinic and not just those enrolled. We have added to the analysis section to note this.

Comment: Also, why are clubs being evaluated with a 3 month timeframe while decentralization uses a 4-month timeframe?

Response: Thank you for catching this typo. It should be 3 months for both. We have made this correction.

Comment: Additional information to understand the rationale for the evaluation of these two interventions may be useful.

Response: These two interventions were the ones that were designated by the NDOH in the Adherence Guidelines for stable patients. The rationale was to offer patients choices for how they received their care, whether in a decentralized medication pick up mode or in a group through a club. To address this, on page 8 we have added the sentence "These two interventions were offered to stable patients in an effort to provide patients with multiple options for where and how to seek care while reducing the congestion in clinics and increasing the convenience for patients."

Reviewer: 2

Reviewer Name: Reed Siemieniuk

Institution and Country: McMaster University, Canada Please state any competing interests: None declared.

Please leave your comments for the authors below

Comment: Thank you for the opportunity to peer review this manuscript. The authors describe their matched-cluster randomised trial that evaluates five different social interventions to improve adherence among people living with HIV in South Africa. Congratulations on the major effort to address important implementation science questions.

Response: We thank the reviewer for the kind words.

Comment: It is obscure to me why the authors have decided to submit the protocol so late. The study has already finished enrolling and follow-up is already finished for most of the patients.

Response: We agree this was an oversight on our part, as we did not realize that BMJ Open published study protocols until late. However, data collection is still ongoing, so we should still meet the requirements.

Comment: There are so few clusters that there is very likely going to be baseline imbalance. The results, whatever they happen to be, are going to be at high risk of bias.

Response: The reviewer is correct and this has also been pointed out by reviewer 1. As we said in response above, while we initially only planned to deal with this through simple statistical adjustment, we have since realized that we will have sufficient site level data to deal with some of this through a difference in differences approach to adjust for baseline imbalances in outcomes. This is because, since we are using routinely collected data, some of which is collected in electronic format, we will be able to identify outcomes for the entire clinic and not just those enrolled. We have added to the analysis section to note this.

Comment: The authors refer to 12-14 month follow-up as "long-term". This is semantics, but no one who is living with HIV indefinitely would consider 1-year follow-up to be long-term or even medium term. I suggest a change in terminology.

Response: We agree with the reviewer and have revised to use the term final outcome as these are our final study outcomes.

Comment: Were the people who reviewed patients for eligibility blinded, or was anything else done to reduce patient selection bias?

Response: We were not able to blind those reviewing the files, but they do not choose the files they are reviewing, this is done centrally according to a computer algorithm. However, as the reviewer notes, this could lead to misclassification problems. We have noted this in the limitation.

Comment: Why did the authors assume a coefficient of variation (ICC) of 0.1? This can be very difficult to predict and should be justified. Was the same ICC used for every analysis? Was any sensitivity analysis performed across a range of ICCs?

Response: We agree with the reviewer that it is hard to know what this will be in advance. We have used this coefficient of variation in other health related studies and found it to be sufficient. We did not do any sensitivity analyses, instead we will just have to accept what it is in practice once all the data is collected.

Comment: The study sites were matched. The analysis plan does not currently incorporate this might be more reliable (not to mention increase power) if they do.

Response: As noted in response to the issues above, we will use a difference in differences approach to increase our power, but as the reviewer notes, we should also account for the matching. We have added this to our analysis section.

Comment: The authors say that changes in the outcomes of 10% to 15% are "clinically meaningful" and imply that changes less than 10-15% are not clinically meaningful. How did they decide this threshold?

Response: This was through a consensus of the investigators, we asked what we thought would be meaningful for investment in the interventions. We have added to the text to note this.

VERSION 2 – REVIEW

REVIEWER	Becky Genberg Johns Hopkins Bloomberg School of Public Health, Department of Epidemiology, Baltimore, Maryland, USA.
REVIEW RETURNED	21-Nov-2017

GENERAL COMMENTS	No further comments.
----------------------

REVIEWER	Reed Siemieniuk McMaster University, Canada
REVIEW RETURNED	07-Nov-2017

GENERAL COMMENTS	I have no further concerns. Thank you for addressing all of my comments. Good luck with the study!
---